# Leucocyte-Rich Platelet-Rich Plasma Enhances Fibroblast and Extracellular Matrix Activity: Implications in Wound Healing

**DOI:** 10.3390/ijms21186519

**Published:** 2020-09-06

**Authors:** Jeannie Devereaux, Narges Dargahi, Sarah Fraser, Kulmira Nurgali, Dimitrios Kiatos, Vasso Apostolopoulos

**Affiliations:** 1College of Health and Biomedicine, Victoria University, Melbourne, VIC 3011, Australia; jim.kiatos@vu.edu.au; 2Institute for Health and Sport, Victoria University, Melbourne, VIC 3011, Australia; narges.dargahi@vu.edu.au (N.D.); sarah.fraser@vu.edu.au (S.F.); kulmira.nurgali@vu.edu.au (K.N.)

**Keywords:** platelet-rich plasma, PRP, platelet gel, platelet-rich fibrin, fibroblasts, extracellular matrix, leucocytes, wound

## Abstract

Background: Platelet-rich plasma (PRP) is an autologous blood product that contains a high concentration of platelets and leucocytes, which are fundamental fibroblast proliferation agents. Literature has emerged that offers contradictory findings about leucocytes within PRP. Herein, we elucidated the effects of highly concentrated leucocytes and platelets on human fibroblasts. Methods: Leucocyte-rich, PRP (LR-PRP) and leucocyte-poor, platelet-poor plasma (LP-PPP) were compared to identify their effects on human fibroblasts, including cell proliferation, wound healing and extracellular matrix and adhesion molecule gene expressions. Results: The LR-PRP exhibited 1422.00 ± 317.21 × 10^3^ platelets/µL and 16.36 ± 2.08 × 10^3^ white blood cells/µL whilst the LP-PPP demonstrated lower concentrations of 55.33 ± 10.13 × 10^3^ platelets/µL and 0.8 ± 0.02 × 10^3^ white blood cells/µL. LR-PRP enhanced fibroblast cell proliferation and cell migration, and demonstrated either upregulation or down-regulation gene expression profile of the extracellular matrix and adhesion molecules. Conclusion: LR-PRP has a continuous stimulatory anabolic and ergogenic effect on human fibroblast cells.

## 1. Introduction

Platelet-rich plasma (PRP) has gained encouraging recognition as an accepted protocol for treating injuries; however, research has mainly focused on orthobiologic applications with few studies being performed on fibroblasts and their relation to cutaneous delayed wound healing [1,2,3,4,5,6]. Overcoming delayed wound healing has generated a need to incorporate novel therapies such as PRP [7,8]. PRP involves the preparation of autologous and biologically safe blood products that contain concentrated leucocytes and platelets, which releases growth factors, cytokines and chemokines and subsequently influences tissue regenerative cells such as fibroblasts [9,10]. However, the leucocytes within PRP have gained recent adverse attention [11,12] with an emphasis on their contribution to inflammation and premature apoptosis. Therefore, some researchers consider leucocyte-rich, platelet-rich plasma (LR-PRP) to be potentially harmful to tissue regeneration [13,14,15]. This study engaged in three in vitro experiments to elucidate the effects of LR-PRP on challenged fibroblasts, as follows: (i) fibroblast proliferation assay; (ii) scratch wound assay; and (iii) the human extracellular matrix (ECM) and adhesion molecules RT^2^ Profiler PCR gene array.

The LR-PRP is obtained by density gradient centrifugation of whole blood, which separates the plasma, leucocytes and platelets from the red blood cells (RBCs) to form a buffy coat. The buffy coat is resuspended into a small amount of plasma to form the final LR-PRP product. Leucocyte-poor, platelet-poor plasma (LP-PPP) is the residual plasma once the LR-PRP is extracted [16]. Plasma subsequently conserves and transports platelets, leucocytes and nutrients [17,18]. The leucocytic fraction of PRP is often uncharacterized with the imprecise umbrella term “leucocyte-rich” employed [19]. For clarity, this study defines LR-PRP as containing a white blood cell (WBC) concentration greater than 100% of that of whole blood and LP-PRP as containing a WBC concentration less than 100% of that of whole blood [11]. In order to simulate a fibroblast in the edges of a chronic wound the fibroblasts were deprived of nutrients by being placed in serum-free media. Under these conditions, fibroblasts conserve energy and cease proliferation [20,21,22]. The challenged fibroblasts were supplemented with LR-PRP and LP-PPP, and the effects were evaluated. 

Fibroblasts were chosen as the experimental cells in this study due to their critical role in wound healing [23,24]. Effective wound healing relies on fibroblast differentiation into myofibroblasts, and collagen type I and III fibril accumulation [25]. In compromised wound healing, collagen fibrils fragment, inhibiting fibroblasts interactions, wound contraction and ECM remodelling [7,23,26]. The ECM is a three-dimensional multi-molecular network that defines a tissue’s shape, structure and functions. ECM remodelling is mediated by an elaborate network of proteinases (e.g.; matrix metallopeptidase [MMP]) and adhesion proteins, which are secreted from stromal cells, such as fibroblasts, epithelial and endothelial cells. MMPs are by-products of fibroblasts, macrophages, and migrating keratinocytes that modulate cell surface receptors, cell signalling, cell death, inflammation and will spontaneously inactivate once the wound has closed [27,28]. Moreover, MMPs disassemble an impaired ECM, regulate extracellular tissue signalling networks, and induce angiogenesis and immune responses [29]. Alternatively, adhesion molecules facilitate the reassembly of the ECM via cell-to-cell and cell-to-ECM cross-talk [30]. Cell adhesion molecule include the integrins (*ITGs*), selectins, intercellular adhesion molecules (*ICAM*), neural cell adhesion molecules (*NCAM*), vascular cell adhesion molecules (*VCAM*), and the catenins [30,31]. These molecules, in conjunction with growth factors and leucocytes, generate an ideally organised ECM infrastructure, which synchronises cell growth and optimises wound healing [32,33,34]. 

Tissue regeneration in wounds relies on leucocytes [8]. Some of the monocytes in wounds polarize into anti-inflammatory and reparative M2 macrophages [35,36,37,38]. Additionally, platelets release a multitude of growth factors, such as platelet derived growth factor, vascular endothelial growth factor and transforming growth factor (TGF-β) [39,40], the latter promoting fibroblast proliferation and recruitment to the wound bed [41,42]. Consequently, LR-PRP which contains both WBCs and platelets, probably could serve as a complementary treatment for delayed woundhealing.

## 2. Results

### 2.1. Haematological Values: Whole Blood, LR-PRP and LP-PPP

The haematological data for the whole blood, LR-PRP and LP-PPP used are available in Appendix A. Haematocrit percentages of the initial collected blood, LR-PRP and LP-PPP volume were calculated by applying the formulae that are available in Appendix A [18,43].

### 2.2. LR-PRP and LP-PPP Enhance Proliferation of Fibroblast Cells

The 3-(4,5-dimethylthiazol-2-yl)-2,5-diphenyl tetrazolium bromide, a tetrazole) (MTT) assay was performed to evaluate the influence of LR-PRP and LP-PPP on fibroblast proliferation at 4, 8, 12, 24 and 48 h (h) time points (Figure 1). At 4 h, LR-PRP proliferation was significantly higher than the negative control (** *p* = 0.003). At 8 h LP-PPP proliferation was significantly increased compared to the negative control (* *p* = 0.024). At 12 h, LR-PRP proliferation was highly significantly different in comparison to the negative control (*** *p* < 0.001). The 12 h was the elected time period for the analysis of the expression of mRNA genes of human ECM and adhesion molecule for this study.

### 2.3. LR-PRP and LP-PPP Enhance Fibroblast Migration in Wound Scratch Assay

LR-PRP and LP-PPP were applied to challenged fibroblasts to assess their effects on migration, via an in vitro scratch wound assay. The migration rate was measured by the shortest wound edge distance of each scratch. The fibroblast cell monolayer migration for wound closure time and migration rate (nm/h) were calculated. At time 8 h and 12 h, there were no significant difference in wound closure compared to the negative control. At the 24 h time point LR-PRP showed approximately 50% wound closure and was significantly increased compared to the LP-PPP (* *p* = 0.033). Wells supplemented with LR-PRP achieved 100% wound closure at 48 h (*** *p* < 0.001), with a migration rate of 20 nm/h. Fibroblast wells supplemented with LP-PPP demonstrated a 60.1% wound closure and a migration rate of 12 nm/h which was similar to the negative control of 63.1% at a rate of 7.3 nm/h at 48 h (Figure 2).

### 2.4. Expression of mRNA Genes of Human ECM and Adhesion Molecules

QIAseq targeted RNA virtual panels were used, in which 84 genes that encode human ECM and adhesion proteins were evaluated. The sample management consisted of LR-PRP versus LP-PPP (control) (Figure 3A), and LR-PRP versus serum-free media (negative control) (Figure 3B). The RNA concentrations and purity were determined by measuring the absorbance by spectrophotometry (Nanodrop, Denovix). The results demonstrated the 260/280 nm and 260/230nm measurements were >1.8 and therefore all samples met the criterion. The RNA integrity number (RIN) was measured via a bioanalyzer (Agilent) in which >7.4 was required for reliable data. In this experiment, the RIN achieved was between 9.3 and 9.7. There was no smearing of the RNA bands, indicating no RNA sample degradation. Furthermore, conforming to the Qiagen software, all fold-changes equal or greater than 2 indicated gene up-regulation. Fold-changes equal or less than minus 2 indicates gene down-regulation.

Notably, 13 gene expressions with significant variances were identified in the LR-PRP and LP-PPP test groups. The genes were labelled by their approved symbols [44], which are numerically classified according to their substrate specificities [45,46,47]. The PCR results for the LR-PRP versus LP-PPP (control) exhibited an up-regulated expression of seven genes: *MMP1*, *MMP3, MMP9, MMP12, ITGA2, ITGA6* and *TIMP3*. Simultaneously, the down-regulation of six genes occurred such as *VCAM1, NCAM1*, *ITGA8, ITGB4, MMP11 and ITGAM*, which is displayed in Figure 3A. In the LR-PRP versus serum-free media (negative control), the up- regulated expression of five genes occurred: namely *MMP1, MMP3, MMP9, MMP12,* and *ITGA2*; in addition to the down-regulation of *MMP11, ITGB4,* and *ITGAM* which is displayed in Figure 3B. The LP-PPP sample versus serum-free media (negative control) exhibited the up-regulation of only *ITGA8*. All 84 genes of human ECM and adhesion molecules are described in Appendix A.

## 3. Discussion

Each of the components in PRP preparations confers specific biologic properties. In clinical and laboratory settings, PRP point-of-care devices vary from commonplace laboratory tubes to highly advanced, automated cell separating apparatuses. The efficiency of platelet recovery fluctuates between systems and subsequently not all PRP formulations are equal [48]. The problem is compounded when publications have not included PRP composition data [42,49,50,51]. To avoid these ambiguities, this study implemented the DEPA classification method to characterize the LR-PRP and LP-PPP [18]. The DEPA classification includes four parameters: dose of platelets; efficiency of PRP production; the purity of PRP, which refers to the relative composition of platelets, leucocytes and RBCs; and the activation method used. Each of these parameters provides information that assists in the interpretation of experimental data [9,52,53].

As reported in Appendix A, the LR-PRP platelet yield and composition indicate high production efficiency. Moreover, the platelet factor increase measure was high (8.75 ± 1.92). The purity was also high as demonstrated by the low WBC and RBC composition of 1.19 ± 0.16% and 0.03 ± 0.01% respectively. Determining the number of RBCs in each sample was important, as clinicians are concerned about their release of reactive oxygen species and the subsequent adverse effects on tissue cells [43,53,54]. The objective of this study was to evaluate the effects of LR-PRP and LP-PPP on human fibroblast activities related to cutaneous wound healing therefore the WBC count was important. The LR-PRP produced attained a white blood cell factor increase of 3.48 ± 0.45 and therefore its designation as LR-PRP was confirmed [11]. Hence, the LR-PRP data attained a DEPA score of AAA.

The effects of leucocytes in PRP has gained recent attention [11,12,55,56] with an emphasis on their influence on inflammation and premature apoptosis [13,14,15]. However, leucocytes comprise only approximately 0.1% to 1% of the final content of PRP. One study argued that LR-PRP may intensify the over-expression of *MMP* and consequently activate ECM catabolic pathways and excessive inflammation [57]. This concept was also accentuated in another study [15], which reported that leucocytes in PRP initiated a greater activation of the nuclear factor kappa-light-chain-enhancer of activated B cells pathway, and subsequently resulting in significantly less fibroblast proliferation and a higher concentration of pro-inflammatory cytokines. However, this study’s PRP contained a high concentration of RBCs and therefore could be the source of this reaction. Dohan Ehrenfest et.al [58] demonstrated that LR-PRP generated a more sustainable fibrin matrix compared to leucocyte-poor PRP, which directly challenges the previous findings [15]. To our knowledge, no clear evidence has confirmed that leucocytes initiate catabolic pathways or that catabolic pathways are even related to leucocyte concentrations. In fact, some studies have illustrated that leucocytes cause no adverse effects and moreover have demonstrated that wound healing essentially relies upon immune-modulatory mechanisms [3,14,59,60,61,62]. For instance, peripheral blood monocytes propagate pluripotent stem cells, which are subsequently induced by the macrophage colony-stimulating factor. Monocytes also transform into anti-fibrotic keratinocyte-like cells and therefore demonstrate regenerative effects [63,64,65]. Additionally, infection is a critical risk factor in chronic wounds, and therefore leucocytes introduced into the wound, via LR-PRP, would exert an antimicrobial effect [66].

Clearly, the role of leucocytes within the wound milieu has not been fully elucidated. The aim of this study’s three experiments was to explore the effect of leucocytes contained in LR-PRP on a key cell that plays a central role in wound healing namely the fibroblast. In the proliferation experiment, the aim was to determine the effects of LR-PRP on fibroblast which were challenged in serum-free media and consequently ceased proliferation to conserve energy [21,22,67]. This assay identified the optimal fibroblast RNA extraction time to be 12 h; however the time limit of 48 h was insufficient to produce fibroblast proliferation data. For instance, previous studies allowed extended time frames of up to 7 days and subsequently only reaching significant proliferation at day 3 and 5 [68,69,70,71]. In the scratch wound assay, LR-PRP achieved 100% closure within 48 h and also accomplished the greater speed of 20 nm/h compared with the LP-PPP of 12 nm/h and the negative control at 7.3 nm/h. According to these data, we can positively propose that leucocytes, and likewise platelets, demonstrate a continuous stimulatory anabolic and ergogenic pathway in tissue repair, rather than a catabolic pathway as proposed in some of the literature [15].

To our knowledge the human RT^2^ Profiler PCR gene array data provide one of the first studies into LR-PRP and the human ECM and adhesion molecules such as transmembrane receptors, E-cadherin, basement membrane constituents, collagen and ECM structural constituents, proteases and protease inhibitors. The ECM is of a major interest in wound therapy because the outcome of the tissue’s architecture and anatomical function is determined during the ECM turnover processes, thus its disassembly and reassembly [72]. These ECM turnover processes are sustained via the facilitation of proteolytic enzymes such as MMPs. MMPs are by-products of fibroblasts, macrophages, and migrating keratinocytes that modulate cell surface receptors, cell–cell and cell–matrix signalling, cell death and inflammation in wound healing and will spontaneously inactivate once the wound has closed [27]. This study confirmed the associations between MMPs and the ECM remodeling. For example, the primary functions of *MMP1, MMP3, MMP9* and *MMP12* (up-regulated by LR-PRP) are to orchestrate the disassembly and assembly of the ECM and subsequently provide a conduit for fibroblast migration. In particular, up-regulated *MMP1* degrades collagen type I and III and subsequently facilitates progenitor cell migration. Conversely, the down-regulation of *MMP1* may lead to keloid and hypertrophic scarring [73,74]. Likewise, *MMP3* degrades collagen types I, III, IV, IX and X; proteoglycans; fibronectins; laminin; and elastin during the ECM remodeling phase. Additionally, *MMP*3 regulates transcription and activation of pro-*MMP* (*MMP1*, *MMP7* and *MMP9*), and thus, collagenases, gelatinase B and matrilysin which facilitates ECM turnover. *MMP9* activates and inactivates cytokines and chemokines, recruits endothelial cells, and promotes angiogenesis [75], whereas *MMP12* disassembles the ECM and is significantly elastolytic [29,76]. These findings are consistent with a similar study [27] which observed *MMP1* up-regulation plus exhibited no cell inhibition or cell death after PRP supplementation of fibroblasts. Our results are also in accordance with Cipriani et al. [77] who noted increased expression of *MMP9* in keratinocytes after PRP supplementation, and demonstrated no cell inhibition or cell death. Likewise, our results agree with a study that demonstrated *MMP2, MMP3, and MMP9* up-regulation for up to 6 days in LR-PRP supplemented cells [73]. In this study no cell inhibition or death occurred, which is likely attributed to the regulation of *MMP* activity at the transcription and activation phases coordinated via the *TIMP3*’s (up-regulated by LR-PRP) anti-inflammatory actions and subsequent inhibition of downstream signalling. Moreover, *TIMP3* minimizes the degree of remaining fibrotic tissue [29,78], and therefore coincides with the down-regulation of *MMP11* (in LR-PRP) which if up-regulated would cause excessive catabolic processes and ageing fibroblasts [26]. According to these data, we infer that the MMPs expressed from LR-PRP supplementation facilitates ECM disassembly, and subsequently provides the foundation for the ECM reassembly.

In order to initiate ECM reassembly the cell adhesion molecules, such as integrins, facilitate the anchoring of cells to the ECM through the binding of fibronectin, vitronectin and collagen [33]. Adhesion molecules provides a mechanical connection between the ECM and the cytoskeleton, and functions as two-way signal-transducing receptors, linking the intracellular and extracellular environments [79]. This enhances growth factor signalling by regulation of the number and stability of receptor tyrosine kinases on the plasma membrane [80,81]. Additionally, directional cell migration within wounds relies on integrin endocytosis and trafficking to coordinate the assembly of new adhesions and their subsequent dissolution. Likewise, maximum cell migration speed is determined by a medium strength cell adhesion, which is facilitated by integrin expressions and ligand binding affinity [81]. The up-regulation of *ITGA* via the LR-PRP was demonstrated by the enhanced directional cell migration and rate of speed in the wound scratch assay. Hence, the LR-PRP achieved 100% wound closure and a significantly faster closure speed compared with the LP-PPP. Moreover, LR-PRP up-regulated *ITGA2*, which initiates collagen synthesis, as well as regulating inflammation and smooth muscle cell proliferation. This is coupled with the up-regulation of *ITGA6* via LR-PRP, which promotes the recruitment of human mesenchymal stem cells to wounded tissue, and the assembly of hemidesmosomes in aide of keratinocyte migration [31]. Hence, *ITGA6*’s profound mitogenic effect on stem cell proliferation could anticipate LR-PRP as a less expensive and lower risk substitute to the harvesting of stem cells [82]. Moreover, *ITGB4*, which is down-regulated via LR-PRP indicates the presence of fibroblast growth factors released from fibroblasts by the stimulation of the LR-PRP. Conversely, if levels of *ITGB4* are increased, which is abnormal and caused by a variety of cellular insults, this causes a deprivation of growth factors and pro-apoptosis expression. Hence, the LR-PRP exhibits a homeostatic fibroblast activity [83]. Additionally, in regular wound healing, the ratio between *ITGB4* and *ITGA6* is instinctively balanced as a means to promote growth factor optimization and moderate TGF-β signalling, which is critical for cutaneous regeneration [31]. As a response to wound stress, however, TGF-β will excite quiescent fibroblasts to differentiate disproportionately into myofibroblasts, resulting in excessive ECM deposition and therefore fibrosis [84]. In the LP-PPP versus negative control, *ITGA8* (responsible for TGF proliferation) was the sole up-regulated gene with no accompanying down-regulated gene expressions [85]. The absence of MMPs in this circumstance would ensure a compromised ECM turnover and subsequently precede hypertrophic or keloid formation [86].

Although some studies proclaim that leucocytes generate catabolic pathways, leucocytes are essentially fundamental to homeostasis especially by eliminating pathogenic and injurious stimuli [10]. For example, macrophages, T lymphocytes, and natural killer cells initiate immune responses via tumor necrosis factor-α release, which subsequently stimulates *VCAM1* and thus is down-regulated via LR-PRP [87]. *VCAM1* expresses on the membrane of activated endothelia in skin, regulates the trans-endothelial migration of leucocytes and leucocyte-endothelial cell adhesion and mediates immunoglobulin genes, mononuclear cells, eosinophils and vascular endothelium adhesion molecules. The up-regulation of *VCAM1* occurs in systemic oxidative stress, myeloperoxidase, autoimmune disease and particularly in malignant melanoma. Hence, the down-regulation of *VCAM1* via the LR-PRP demonstrates homeostatic fibroblast activity [88,89]. Comparably, the leucocyte dependent integrin gene, *ITGAM*, is down-regulated via the LR-PRP, which subsequently reduces leucocyte adhesion and migration, and facilitates phagocytosis. The benefits of *ITGAM* down-regulation are illustrated in the LR-PRP wound scratch assay, which restored the challenged fibroblasts to normal activity, and no inhibition or cell death occurred. Finally, *NCAM1* (down-regulated via LR-PRP) primarily induces synaptic plasticity in the nervous system; however, it is also present in non-neural tissues. *NCAM1* facilitates fibroblast growth factor receptor signalling [90], cell-matrix adhesion, cell migration and subsequently tissue regeneration [91,92]; however, when up-regulated *NCAM1* serves as an immunohistochemical skin cancer marker [93,94].

Although there is a contrast in LR-PRP and LP-PPP cell concentrations, and it was expected LR-PRP would be significant at all times, the LP-PPP too exhibited significant differences. For instance, no significant changes of *NCAM1* and *VCAM1* occurred in the LR-PRP versus serum-free (negative control), thus indicating LP-PPP possibly contributed to their enhancement. This result may be explained by the fact that plasma, which contains albumins, globulins, fibrinogen, hormones, electrolytes, glucose, gases and IGF-1, shares parallel importance to platelets and leucocytes in wound healing [17]. Previous studies have shown plasma stimulates procollagen type 1, protein production and *MMP1* and *MMP3* expressions; although the PRP was significantly higher [70,95]. The researcher’s focus on platelet concentration efficacy in PRP overlooked the properties of plasma and may have underestimated its value.

This study did not include in vivo experiments and therefore represents a scientific understanding of LR-PRP in vitro. Subsequently these findings cannot be extrapolated directly into clinical applications as each patient’s condition requires careful consideration. The scope of this study was focused on the expression of genes after 12 h of incubation in order to determine the changes early on. However, it would be of interest to determine gene expressions at different time points such as, 48, and 72 h, and to determine if these significant changes are transient or continuous between the LR-PRP and LP-PPP. Likewise, future experiments should include chemokines, cytokines, and various genes at different time points, namely 0, 4, 8, 12, 24, 48, and 72 h, to provide insights into which components are being expressed or consumed and secreted by cells in response to LR-PRP and LP-PPP. As previously mentioned, LR-PRP and LP-PRP, in which the numbers of platelets are similar would clarify the effects of leukocytes more so than comparing LR-PRP and LP-PPP. Further studies that adopt these recommendations would contribute to the development of targeted interventions aimed at ECM diseases.

## 4. Materials and Methods

### 4.1. Samples

Whole blood, LR-PRP and LP-PPP residual samples were donated, to be used in these experiments, by six females aged between 31 to 59 years undergoing an elective LR-PRP treatment unrelated to this study. The donors complied with the physician’s health suitability criteria which excluded patients with conditions listed in Appendix A. Donors were referenced by assigned numbers by the treating clinician. This research was approved by the Victoria University Human Ethics Committee (Approval No. HRE18-041, Thursday, May 3rd, 2018).

### 4.2. Analysis of Leucocyte-Rich, Platelet-Rich Plasma and Leucocyte-Poor, Platelet-Poor Plasma

The Emcyte^®^ PurePRP II manufacturer’s instructions were adhered to. Isolation, storage and delivery of the blood samples were carried out according to the Good Manufacturing Practice Guidelines and the Australian Regulatory Guidelines for Medical Devices (Therapeutics Goods Administration Guideline, 2018). Samples of whole blood, LP-PPP and LR-PRP were immediately analysed by the Dorevitch Pathology Clinical Trials laboratory (Western Health, Footscray Hospital, Melbourne, VIC, Australia) for a complete data determination. The original 50 mL of whole blood was blended with 10 mL of sodium citrate, which equally diluted the whole blood, LR-PRP, and LP-PPP. For rigorous reporting of data, the DEPA classification method was used to characterize all samples, which is based on biological parameters that are classically used in the cell therapy field (described in Appendix A) [18,43]. After the cell calculations were completed, the LR-PRP and LP-PPP were activated using 10% CaCl^2^ at a volume ratio of 10:1 and then were incubated at 37 °C and 5% carbon dioxide for 1 h. The LR-PRP and LP-PPP were centrifuged at 1900 g Force for 10 min to separate any clots. The LR-PRP and LP-PPP were stored at −20 °C until required.

### 4.3. Cell Culture

The human caucasian foetal foreskin fibroblast cell line (HFFF2) (Cell Bank Australia) was used in this study. The fibroblast cells were cultured in Dulbecco’s modified Eagle medium GlutaMAX™-1 media and supplemented with 100 U/mL penicillin and 100 µg/mL streptomycin (Gibco, Melbourne, Australia) at 37 °C and 5% carbon dioxide. Fibroblasts were expanded and at the seventh passage, all multiple cells in cryovials were frozen and stored in liquid nitrogen (cell bank) before being used.

### 4.4. Fibroblast Proliferation Method Using MTT

The effects of 5% LR-PRP and LP-PPP on human dermal fibroblasts were compared via a colorimetric assay for assessing cell metabolic activity, namely MTT [96]. The 5% dose was established according to the results of previous studies [68,70]. The MTT assay was performed on *n* = 3 samples, and for each sample 6 replicate wells were used. Fibroblasts were seeded at a density of 1 × 10^3^ cells/well in 96 U-shaped well culture plates and then incubated for 48 h to allow for adhesion in complete medium of up to 80% confluence in order to prevent an artifactual decrease in proliferation. The complete media was aspirated and replaced with serum-free media for 24 h in order to simulate cell fibroblasts in chronic wounds [20]. The fibroblasts cells were evaluated as follows: (1) 5% LR-PRP; (2) 5% LP-PPP; (3) serum-free media (negative control). Following intervention, the cells were incubated for 4, 8, 12, 24 and 48 h. The absorbance of each sample was measured at each time point at the optical density of 570 nm using the BIO-RAD iMark microplate reader.

### 4.5. In Vitro Scratch Wound Assay

The in vitro scratch wound assay was utilized to determine directional cell migration based on the observation of cell mobility into a scratch “wound” created in the fibroblast cell monolayer [68]. The assay was performed on *n* = 3 samples, and 6 replicate wells were used for each sample. Human fibroblasts were seeded to a final density of 3 × 10^4^ cells/well in 24-well microplates with 500 µL of complete media and incubated for 48 h to produce a maximally confluent monolayer. The fibroblast samples were evaluated as follows: (1) 5% LR-PRP; (2) 5% LP-PPP; and (3) serum-free media (negative control). After each scratch was created, the cells were washed with PBS. To measure the cell mobility, the status of the scratch wound was monitored via the time lapse imaging system BioStation IM-Q. Images were collected at time points of 0, 4, 8, 12, 24 and 48 h. All images were uploaded into ImageJ (NIH, Bethesda, MD, USA), and their size was equally adjusted. A global scale was set from a reference micrometre/micro-ruler image. The time of the image and length of the wound were measured. The magnitude of the wound closure was measured and plotted against time. The decreases of the scratch width over the 48 h period were calculated as the average distance between the wound edges in µm. The wound area was calculated by tracing the cell-free area in captured images. Wound closure and cell migration rates were calculated according to Grada, et al. [97].

Wound closure percentage (%) is the measurement of the fibroblast migration rate expressed as the change in the wound area over 48 h divided by the change in wound width by the time spent in migration.
(1)$$\mathrm{Wound}\;\mathrm{Closure}\;\%\;=\;[\frac{A_{t\;=\;0h\;}-\;A_{t\;=\;}△h}{A_{t\;=\;0h}}]\;\times \;100\%$$

$A_{t\;=\;0\;h\;}$ is the area of the wound measured immediately after scratching (*t* = 0 h); $A_{t=}△h$ is the area of the wound measured hours after the scratch is performed. The cell migration rate equals the change in wound width or closure divided by the time spent in migration; the closure percentage is expected to increase as the cells migrated over the 48 h.

Hence, the Cell Migration formula is:(2)$$R_{M}\;=\;(W^{i\;}-W^{f})\;÷\;t$$

($R_{M})$ = rate of cell migration (nm/h); $W_{i\;}$= initial wound width (nm); $W_{f}$ = final wound width (nm); *t* = duration of migration in hours.

### 4.6. The Human ECM and Adhesion Molecules RT^2^ Profiler PCR Gene Array

The expression of genes was elucidated using the RT2 Profiler PCR Array for 84 human ECM and cell adhesion genes (Qiagen, Chadstone, VIC, Australia). This array detects gene expressions that encode human ECM proteins, including transmembrane receptors, adhesion molecules, E-cadherin, ECM adhesion, basement membrane constituents, collagen, ECM structural constituents, ECM proteases and protease inhibitors. The array contains a panel of 5 housekeeping genes used for normalisation of data, plus a panel of three reverse transcription controls, a genomic DNA control and three positive PCR controls.

In this experiment, human fibroblasts cells were seeded to a density of 3 × 10^6^ cells in a 40 mL flask containing 12 mL of complete media and incubated for 48 h to produce a confluent monolayer. The fibroblasts were supplemented as follows: (1) 5% LR-PRP; (2) 5% LP-PPP; and (3) serum-free media (negative control). Following 12 h incubation, fibroblast cells were extracted by standard laboratory enzymatic methods [98]. Each cell pellet was individually collected for fibroblast mRNA extraction using the RNAeasy mini kit (Qiagen) exactly according to the manufacturer’s instructions. Thereafter, RNA samples were used to assess the RNA quality and concentration using Nanodrop (Denovix, Melbourne, Australia). The remaining RNA were aliquoted (6 µL RNA) into individual 0.5 mL micro-tubes for mRNA expression analysis according to the method previously described by Dargahi [99].

### 4.7. Data Analysis Using Prism Programs and Qiagen Software Programs

The statistical analyses for the fibroblast proliferation and scratch wound assay were performed using GraphPad Prism software, version 7.0e. Data were tested for normality and equal variance before analysis. The 2-way ANOVA and Tukey’s multiple comparison tests were used with values of *p* < 0.05 being regarded as significant. Data are presented as mean ± SEM. To analyse the human ECM and adhesion molecules RT^2^ Profiler PCR array the Qiagen software was used based on the −$\Delta \Delta C_{T}\;$ method with normalisation of the raw data to 5 housekeeping genes, 1 DNA control, 3 reverse transcription controls and 3 + PCR controls. Fold-change (2^ (−$\Delta \Delta C_{T}$)) is the normalized gene expression (2^(−$\Delta C_{T}$)) in the test sample divided by the normalized gene expression (2^ (−$\Delta C_{T}$)) in the control sample. Fold-regulation represents fold-change results in a biologically meaningful way. Fold-change values greater than 2 indicate a positive- or an up-regulation, and the fold-regulation is equal to the fold-change. Fold-change values less than minus 2 indicate a negative or down-regulation, and the fold-regulation is the negative inverse of the fold-change. The p values are calculated based on a Student’s t-test of the replicate 2^ (−$\Delta C_{T}$) values for each gene in the control group and treatment groups.

## 5. Conclusions

The present study provides indications of the ability of LR-PRP and LP-PPP to induce challenged fibroblasts in vitro. This study did not include in vivo experiments and therefore may not be interpreted to clinical applications. The fibroblast proliferation and wound scratch assay supported the hypothesis in which the ability of LR-PRP to affect biological responses of fibroblasts did not exhibit catabolic effects. The ECM and adhesion molecule expressions that we identified contribute to our scientific understanding of the interplay and cross-talk between genes in the disassembly and reassembly of the ECM during wound healing. These insights add knowledge to the rapidly expanding field of ECM-driven diseases, particularly in the identification of underlying genetic bases and abnormal pathways. Additionally, our results provide insight into how these genes respond to LR-PRP and, by extension, other related autologous cell therapies. Overall, our findings suggest that high concentrations of leucocytes and platelets demonstrate a stimulatory continuous anabolic and ergogenic pathway on fibroblasts. Therefore, we propose that LR-PRP can be successfully employed to facilitate wound healing.

## Figures and Tables

**Figure 1 ijms-21-06519-f001:**
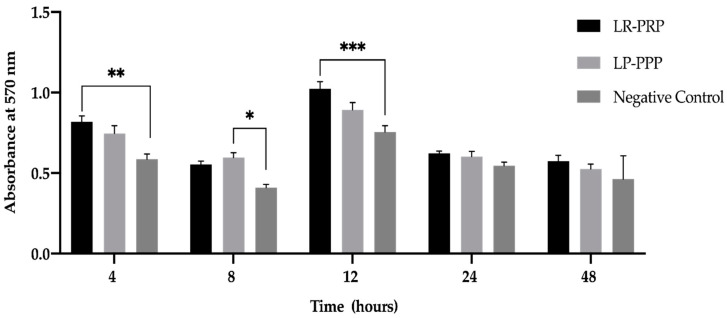
MTT Fibroblast Proliferation Assay. LR-PRP, LP-PPP, serum-free media (negative control) measured at 5 time points 4, 8, 12, 24, 48 h. Data were analyzed and the means for n = 3 samples (6 replicate wells for each sample) were calculated and presented as plus or minus (±) the standard error of the mean (SEM) using a two-way ANOVA and Tukey’s multiple comparison tests. Symbols represent the p value; * *p* < 0.024 significant difference, ** *p* < 0.003 very significant difference, *** *p* < 0.001 highly significant difference. Abbreviations: Leucocyte-rich, platelet-rich plasma (LR-PRP), leucocyte-poor, platelet poor plasma (LP-PPP).

**Figure 2 ijms-21-06519-f002:**
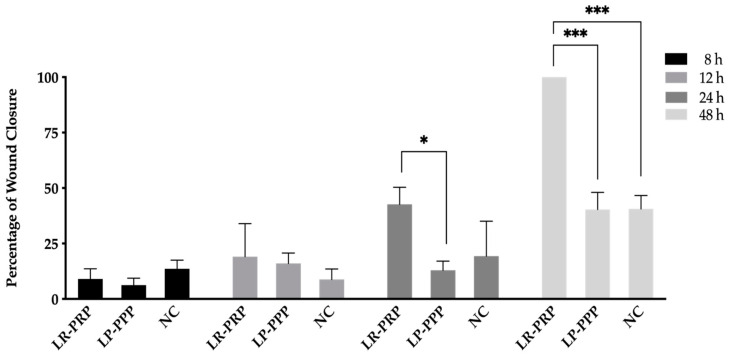
Scratch wound assay. Wound closure for each time point and group is measured in percentages. Data were analyzed for *n* = 3 samples (6 replicate wells for each sample) using the mean ± standard error of the mean (SEM) using a two-way ANOVA and Tukey’s multiple comparison tests (* *p* = 0.033 statistically significant difference, *** *p* < 0.001 very highly statistically significant difference). Abbreviations: Leucocyte-rich-platelet-rich plasma (LR-PRP), leucocyte-poor-platelet poor plasma (LP-PPP), negative control (NC).

**Figure 3 ijms-21-06519-f003:**
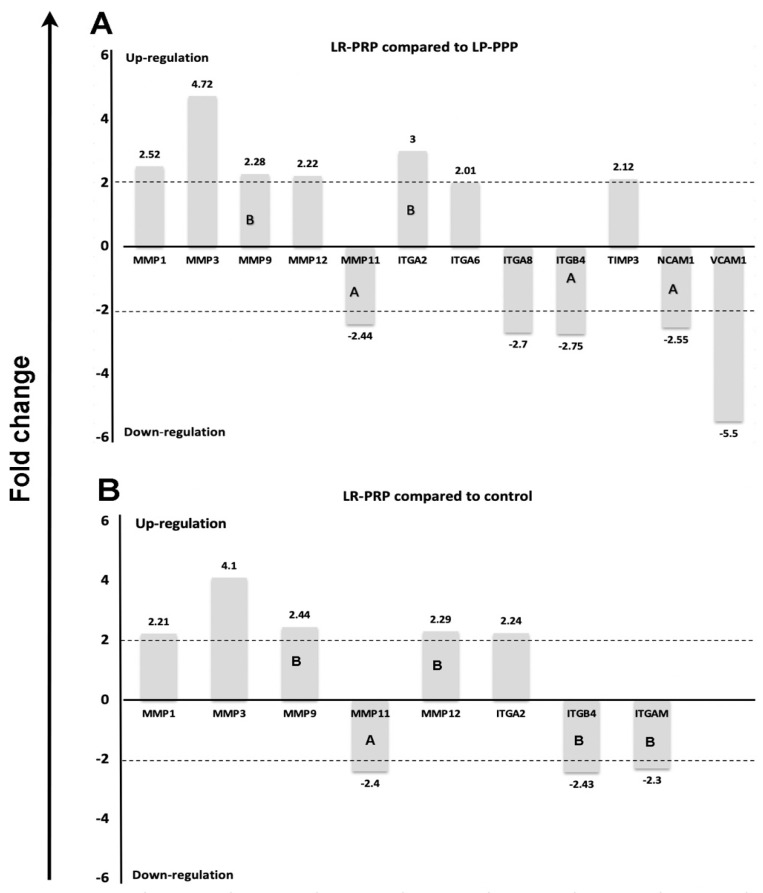
The human extracellular matrix and adhesion molecules RT^2^ Profiler PCR gene array. Fibroblasts supplemented with (**A**) LR-PRP versus LP-PPP (control) at 12 h or (**B**) LR-PRP versus serum-free media (negative control) representing significant up- and down-regulated genes (more than two-fold) and those with no significant change (less than two-fold). Abbreviations: *MMP1*, matrix metallopeptidase 1 (collagenase); *MMP3*, matrix metallopeptidase 3 (stromelysin); *MMP9*, matrix metallopeptidase 9 (gelatinase); *MMP11*, matrix metallopeptidase 11 (stromelysin); *MMP12*, matrix metallopeptidase 12 (metalloelastase); *ITGA2*, integrin alpha 2; *ITGA6*, integrin, alpha 6; *ITGA8*, integrin alpha 8; *ITGAM*, integrin alpha M chain; *ITGB4*, integrin beta 4; *TIMP3*, metalloproteinase inhibitor 3; *NCAM1*, neural cell adhesion molecule 1; *VCAM1*, vascular cell adhesion molecule 1, leucocyte-rich—platelet-rich plasma (LR-PRP), leucocyte-poor—platelet poor plasma (LP-PPP). Letter A specifies that the gene’s average threshold cycle is relatively high (>30) in either the control or the test sample and is reasonably low in the other sample (<30). These data mean that the gene’s expression is relatively low in one sample and reasonably detected in the other sample, which suggests that the actual fold-change value is at least as large as the calculated and reported fold-change result. Letter B specifies that the gene’s average threshold cycle is relatively high (>30), meaning that its relative expression level is low, in both control and test samples, and the *p* value for the fold-change is either unavailable or relatively high (*p* > 0.05).

## Data Availability

The datasets used and/or analyzed during the current study are available from the corresponding author on reasonable request. Patient identifying information is not included in the study. All patient consent forms for publication are on file at Victoria University, Melbourne VIC Australia.

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
