# Peer review of "Leucocyte-Rich Platelet-Rich Plasma Enhances Fibroblast and Extracellular Matrix Activity: Implications in Wound Healing"

_ijms, 2020, doi:10.3390/ijms21186519_

Round 1

Reviewer 1 Report

This manuscript tries to demonstrate how LR-PRP treatment enhance starved fibroblasts on proliferation, migration and gene array analysis in vitro compared with LP-PPP to prove whether LR-PRP treatment is clinically relevant for accelerating wound healing process against injuries w.

Three in vitro assay analysis , MTT assay for proliferation assay, scratch-wound assay, and ECM/adhesion molecules gene array, are straight forward methods and results clearly showed LR-PRP was superior to either LP-PPP or serum-free, serum rich media on MTT and scratch assays whereas there was no sign for cell growth inhibition and apoptosis according to the gene array data.

First of all, although overall results (three sets) and implication from these data is relatively simple, the length of this manuscript is extremely long and too many redundant descriptive statements among results, figure legends, material & methods, and discussion part.

Also Figure 4 should be removed or moved to the supplemental section. I believe this manuscript can be shortened to half of its length or even more.

There are additional several questions raised for experimental design and procedures.

  1. How many individuals for samples were used in MTT assay and wound scratch assay, respectively? In Figure 1 and Figure 2, n=3 and N-4 means three and four different individual samples were used for each experiment? If so, each individual data could be important since PRP treatment is autologous and efficacy varies among treatment as the authors stated in the introduction.

  1. Gene array data performed only at 12hours due to the significant changes in proliferation for LR-PRP (Fig.1) is not appropriate explanation for choosing single time point. The authors should perform multiple time points for this kind of gene expression array analysis, at least 48hrs, and compare these significant changes are transient or continuous since significant difference were observed in scratch assay between LR-PRP and LP-PPP at 48hrs..

  1. In Fig.3 A (LP-PPP vs LR-PRP) and Fig.3 B (Serum-free vs LR-PRP), only A panel shows significant down regulation of adhesion molecules MCAM1 and VCAM1 whereas no such significant changes in B indicating LP-PPP may have potential to enhance these molecules rather than down regulation by LR-PRP. Please explain this possibility.

  1. Citation format is not appropriate as there is no need to cite name of all co-authors in the main text. Please fix them.

Author Response

REVIEWER – 1

Point 1:  First of all, although overall results (three sets) and implication from these data is relatively simple, the length of this manuscript is extremely long and too many redundant descriptive statements among results, figure legends, material & methods, and discussion part.

Response: Thank you for your comments. After adding in the other reviewers changes we summarised results, figure legends, materials and methods and reduced the article by 70 lines. We also moved some figures and tables to supplementary materials.

Point 2: Figure 4 should be removed or moved to the supplemental section. I believe this manuscript can be shortened to half of its length or even more.

Response: Figure 4 and Tables A1 and A2 are now in the supplemental section. The article has been shortened as suggested. 

Point 3: How many individuals for samples were used in MTT assay and wound scratch assay, respectively? In Figure 1 and Figure 2, n=3 and N-4 means three and four different individual samples were used for each experiment? If so, each individual data could be important since PRP treatment is autologous and efficacy varies among treatment as the authors stated in the introduction

Response: We thank the reviewer for the comment. We have corrected this and changes have been applied to Figures 1 and 2 and in the methods -  line 337 to 364.

Point 4: Gene array data performed only at 12hrs due to the significant changes in proliferation for LR-PRP (Fig.1) is not appropriate explanation for choosing single time point. The authors should perform multiple time points for this kind of gene expression array analysis, at least 48hrs, and compare these significant changes are transient or continuous since significant difference were observed in scratch assay between LR-PRP and LP-PPP.

Response:  We appreciate this insight and have highlighted this limitation in the conclusion and provided future recommendations according to the reviewers advice.

Point 5: In Fig.3 A (LP-PPP vs LR-PRP) and Fig.3 B (Serum-free vs LR-PRP), only A panel shows significant down regulation of adhesion molecules NCAM1 and VCAM1 whereas no such significant changes in B indicating LP-PPP may have potential to enhance these molecules rather than down regulation by LR-PRP. Please explain this possibility

Response:  We thank the reviewer for this important comment. This insight has now been highlighted in lines 299-308.

Point 6: Citation format is not appropriate as there is no need to cite name of all co-authors in the main text. Please fix them

Response:  All in-text references have been updated to comply with IJMS referencing style.

Reviewer 2 Report

I read with interest the manuscript by Devereaux and collaborators entitled "Leucocyte-rich, platelet-rich plasma enhances fibroblast and extracellular matrix activity: implications in wound healing".

The authors conducted a study to investigate the effect of the treatment with plasma enriched with leukocytes and platelets (LR-PRP) on the wound repair process. The study was conducted in vitro by performing cell viability (MTT) and cell migration (wound-healing) assays on fibroblast cultures (HFFF2) after treatment with LR-PRP or LP-PPP (well characterized in terms of blood cells composition) and using untreated cells as negative control. Furthermore, the authors investigated the gene expression changes of a panel of 84 genes involved in extracellular matrix (ECM) remodeling.

The authors infer from the obtained results that treatment with LR-PRP improve wound repair as it increases fibroblast proliferation, activates cell migration and modulates the expression of genes related to ECM remodeling and cell adhesion.

The study is sound and the manuscript is well written. Introduction is exhaustive and discussion of results is carried out in comparison with the literature data. Materials and methods used for the experiments are well described.

Minor corrections:

1) line 106 - The following sentence should be checked for typo: "For each time point, except for the 24 h time point, there was no statistically significant difference in proliferation between the LR-PRP and the LP-PPP samples (p < 0.05)". In fact, the not statistically significant difference is indicated as p> 0.05.

2) As in "Data Analysis Using Prism Programs and Qiagen Software Programs" (Materials and Methods, paragraph 4.7) the authors clearly explain that the "Fold-changes (2 ^ (- ΔCT)) were calculated using the normalized gene expression (2 ^ (- ΔCT)) in the group sample divided by the normalized gene expression (2 (-ΔCT)) in the control sample" then to avoid misunderstandings in the results description, the following sentences could be corrected inverting the order of the samples in comparison:

- line 144-145 - "The sample management consisted of LP-PPP (control) versus LR-PRP (Figure 3A), and serum-free (negative control) versus LR-PRP (Figure 3B)" could be changed to "The sample management consisted of LR-PRP versus LP-PPP (control) (Figure 3A), and LR-PRP versus serum-free (negative control) (Figure 3B)".

- line 158 - "The LP-PPP (negative control) versus LR-PRP expressed an up-regulation of 7 genes" could be changed to "The LR-PRP versus LP-PPP (negative control) expressed an up-regulation of 7 genes".

- line 166 - "In the serum-free (negative control) versus LR-PRP, an up-regulation of 5 genes were expressed" could be changed to "In the LR-PRP versus serum-free (negative control), an up- regulation of 5 genes were expressed".

- line 168 - "The serum-free media (negative control) versus LP-PPP only exhibited the up-regulation of only ITGA8" could be changed to "The LP-PPP versus serum-free media (negative control) exhibited the up-regulation of only ITGA8".

- Figure 3A, title - "LP-PPP vs LR-PRP fold regulation" could be changed to "LR-PRP fold regulation vs LP-PPP".

- Figure 3B, title - "Serum-free (negative control) vs LR-PRP fold regulation" could be changed to "LR-PRP fold regulation vs Serum-free (negative control)".

- Figure 4, title of the upper panel - "LP-PPP vs LR-PRP" could be changed to "LR-PRP vs LP-PPP".

- Figure 4, title of the down panel - "Serum-free Media (negative control) vs LR-PRP fold regulation" could be changed to "LR-PRP fold regulation vs Serum-free Media (negative control)".

The following references should be corrected for typos or because of incomplete:

line 533 - ref 14,

line 537 - ref 16,

line 539 - ref 17,

line 554 - ref 23,

line 556 - ref 24,

line 559 - ref 25,

line 568 - ref 29, it is a book, may be is missed a comma after the title.

line 610 - ref 46,

line 619 - ref 50,

line 639 - ref 58,

line 642 - ref 59,

line 646 - ref 60,

line 706 - ref 83,

line 707 - ref 84,

line 714 - ref 87,

line 726 - ref 92,

line 736 - ref 95,

line 738 - ref 96,

line 743 - ref 98,

line 747 - ref 100.

Author Response

REVIEWER - 2

Point 1: The study is sound and the manuscript is well written. Introduction is exhaustive and discussion of results is carried out in comparison with the literature data. Materials and methods used for the experiments are well described

Minor corrections:

Point 2: line 106 - The following sentence should be checked for typo: "For each time point, except for the 24 h time point, there was no statistically significant difference in proliferation between the LR-PRP and the LP-PPP samples (p < 0.05)". In fact, the not statistically significant difference is indicated as p> 0.05.

Response: Thank you for this unique find. The author detected a calculation error and reconducted the statistics. Such as, Figure 1 summarizes the results of a 3-(4,5-Dimethylthiazol-2-yl)-2,5-diphenyl tetrazolium bromide, a tetrazole) (MTT) assay, performed to evaluate the influence of LR-PRP and LP-PPP on fibroblast proliferation at time points of 4, 8, 12, 24 and 48 h. At 4 h LR-PRP was significantly different to the serum-free (negative control), **p = 0.003. At 8 h LP-PPP was significant to the serum-free (negative control), *p = 0.024. At 12 h LR-PRP was highly significantly different to the serum-free (negative control), ***p < 0.001.”

Point 3: As in "Data Analysis Using Prism Programs and Qiagen Software Programs" (Materials and Methods, paragraph 4.7) the authors clearly explain that the "Fold-changes (2 ^ (- ΔCT)) were calculated using the normalized gene expression (2 ^ (- ΔCT)) in the group sample divided by the normalized gene expression (2 (-ΔCT)) in the control sample" then to avoid misunderstandings in the results description, the following sentences could be corrected inverting the order of the samples in comparison: - line 144-145 - "The sample management consisted of LP-PPP (control) versus LR-PRP (Figure 3A), and serum-free (negative control) versus LR-PRP (Figure 3B)" could be changed to "The sample management consisted of LR-PRP versus LP-PPP (control) (Figure 3A), and LR-PRP versus serum-free (negative control) (Figure 3B)".

Response:   Correction to this sentence is in lines 390-403

Point 4: line 158 – “The LP-PPP (negative control) versus LR-PRP expressed an up-regulation of 7 genes” could be changed to “The LR-PRP versus LP-PPP (negative control) expressed an up-regulation of 7 genes”.

Response: Correction made which is now lines 134-135

Point 5: line 166 – “In the serum-free (negative control) versus LR-PRP, an up-regulation of 5 genes were expressed” could be changed to “In the LR-PRP versus serum-free (negative control), an up- regulation of 5 genes were expressed”.

Response: Corrected now in lines 137-138

Point 6: line 168 – “The serum-free media (negative control) versus LP-PPP only exhibited the up-regulation of only ITGA8” could be changed to “The LP-PPP versus serum-free media (negative control) exhibited the up-regulation of only ITGA8”.

Response: Corrected now lines 140-141

Point 7: Figure 3A, title - "LP-PPP vs LR-PRP fold regulation" could be changed to "LR-PRP fold regulation vs LP-PPP".

Response: Title changes have been made.

Point 8: Figure 3B, title – “Serum-free (negative control) vs LR-PRP fold regulation” could be changed to “LR-PRP fold regulation vs Serum-free (negative control)”.

Response: Title changes have been made.

Point 8: Figure 4, title of the upper panel - "LP-PPP vs LR-PRP" could be changed to "LR-PRP vs LP-PPP".

Response: Corrected, Figure 4 is in Supplementary Materials

Point 9: Figure 4, title of the down panel - "Serum-free Media (negative control) vs LR-PRP fold regulation" could be changed to "LR-PRP fold regulation vs Serum-free Media (negative control)".

Response: Corrected to match your previous review of  LP-PPP vs LR-PRP" could be changed to "LR-PRP vs LP-PPP".

Point 10:  The following references should be corrected for typos or because of incomplete:

Response: All references in Endnote were updated and are in accordance to MDPI-IJMS style

Reviewer 3 Report

In this paper, the authors compared leukocyte-rich, platelet-rich plasma (LR-PRP) and leukocyte-poor, platelet-poor plasma (LP-PPP) to identify their in vitro effects on fibroblasts. They showed that LR-PRP was superior in cell proliferation, wound scratch assay, and some gene expressions. Although their results were clear, the results are easily expected and provide little clinical implication. Some concerns are listed below.

Major concerns

1: I cannot understand why the authors compared LR-PRP and LP-PPP. LP-PPP is not used in a clinical situation. Furthermore, the results are easily expected because cell numbers in LR-PRP are much higher in both leukocytes and platelets. If they compared LR-PRP and LP-PRP, in which the numbers of platelets were similar, it would clarify the effects of leukocytes. In the study design of this paper, however, no clinical implication is not obtained.

2: The authors did not analyze any cytokines in LR-PRP and LP-PPP. Comparison of cytokine concentrations (such as platelet-derived growth factor and fibroblast growth factor) would elucidate underlying mechanisms of the effects of LR-PRP on fibroblasts.

Minor concerns

1: Statistical analysis not only for PCR experiments but also for other experiments should be described in the methods section.

2: In PCR experiments, an endogenous reference gene (usually GAPDH or beta-actin) should be described in the methods section.

3: In MTT assay, there was no significant difference between negative control and positive control. How do the authors explain this result?

Author Response

REVIEWER - 3

In this paper, the authors compared leukocyte-rich, platelet-rich plasma (LR-PRP) and leukocyte-poor, platelet-poor plasma (LP-PPP) to identify their in vitro effects on fibroblasts. They showed that LR-PRP was superior in cell proliferation, wound scratch assay, and some gene expressions. Although their results were clear, the results are easily expected and provide little clinical implication. Some concerns are listed below.

Point 1: I cannot understand why the authors compared LR-PRP and LP-PPP. LP-PPP is not used in a clinical situation. Furthermore, the results are easily expected because cell numbers in LR-PRP are much higher in both leukocytes and platelets. If they compared LR-PRP and LP-PRP, in which the numbers of platelets were similar, it would clarify the effects of leukocytes. In the study design of this paper, however, no clinical implication is not obtained.

Response:  Thank you for highlighting this limitation. We have now clarified this in the paper (lines 295-304 and 407-411). 

Point 2: The authors did not analyze any cytokines in LR-PRP and LP-PPP. Comparison of cytokine concentrations (such as platelet-derived growth factor and fibroblast growth factor) would elucidate underlying mechanisms of the effects of LR-PRP on fibroblasts.

Response: Thank you for highlighting this. This point has been raised in future recommendations within the Conclusion section.

Point 3: Statistical analysis not only for PCR experiments but also for other experiments should be described in the methods section.

Response: All statistical methods are now included in the methods section.

Point 4:  In PCR experiments, an endogenous reference gene (usually GAPDH or beta-actin) should be described in the methods section.

Response:  The addition of house-keeping genes – lines 399 – 401 and 416-418 are now included. The expression of genes coding for ECM and adhesion molecules were elucidated using the RT2 Profiler PCR Array for 84 human ECM and cell adhesion genes (Qiagen, Australia ). This array detects gene expressions that encode human ECM proteins, including transmembrane receptors, cell–cell adhesion molecules, E-cadherin, ECM adhesion, basement membrane constituents, collagen and ECM structural constituents, ECM proteases and protease inhibitors. The array also contains a panel of 5 housekeeping genes used for normalization of data, plus a panel of three reverse transcription controls, a genomic DNA control and three positive PCR controls.

Point 5: In MTT assay, there was no significant difference between negative control and positive control. How do the authors explain this result?

Response: We acknowledged the positive control did not work and have therefore removed this data.

Round 2

Reviewer 1 Report

Comment to the Response for Point 3: I still can not clearly understand what the authors say about the origin of the samples. Material section, at 4.1Samples Line 9, saying “six people”, then revised legend for Figure1 saying six replicates/samples; n=3 samples is shown. Does this explanation mean all six individual were used or only three of six donors ?  Or were the samples from six donors mixed altogether?  Same question for the revised Legend for Figure 2.  Also I cannot find corrected sentences between line 337 to 364.

Comment to the Response for Point 4: I cannot find the specific explanation of the limitation for this critical point in revised conclusion.

I also suggest conclusion statement should be more concise and clearer, especially for the limitation of the interpretation of this study to clinical application.

Author Response

Reviewer 1: Comment to the Response for Point 3: I still can not clearly understand what the authors say about the origin of the samples. Material section, at 4.1Samples Line 9, saying “six people”, then revised legend for Figure1 saying six replicates/samples; n=3 samples is shown. Does this explanation mean all six individual were used or only three of six donors ?  Or were the samples from six donors mixed altogether?  Same question for the revised Legend for Figure 2.  Also I cannot find corrected sentences between line 337 to 364.

Response:  Thank you for your comments. The original sample for calculating haematological values is n=6. In the MTT, Wound Scratch Assays sample size is n=3 and for each sample 6 replicate wells were used. For the MTT and PCR experiment the same n=3 was used.

Lines 97, 115

For example:

MTT 96 Well Plate and Wound Scratch Assay 24 Well Plate for Sample1

Wells 

 1

2

3

4

5

6

Sample 1

LR-PRP

LR-PRP

LR-PRP

LR-PRP

LR-PRP

LR-PRP

Sample 1

LP-PPP

LP-PPP

LP-PPP

LP-PPP

LP-PPP

LP-PPP

Sample 1

Negative Control

Negative Control

Negative Control

Negative Control

Negative Control

Negative Control

Reviewer 1 Point 4: 

  1. I cannot find the specific explanation of the limitation for this critical point in revised conclusion.
  2. I also suggest conclusion statement should be more concise and clearer, especially for the limitation of the interpretation of this study to clinical application.

Responses to Point 4:

  1. a) We added a limitation paragraph which has included the highlighted limitation and have also reiterated this limitation in the conclusion – Discussion lines 306-318; Conclusion lines 419-432

  1. b) We have edited the conclusion to be more concise and clearer. We have also included the limitation “of the interpretation of this study to clinical application” - Conclusion lines 419 -432

Reviewer 3 Report

The manuscript has improved after revision.

Some limitations of this study are described in the Conclusion section.  The limitations should be noted in the Discussion section, and the conclusions should be brief and concise.

Author Response

Thank you for your comments

 Reviewer 3 - Point 1: Some limitations of this study are described in the Conclusion section.  a) The limitations should be noted in the Discussion section, b) and the conclusions should be brief and concise.

Response to Point 1

  1. a) All limitations have been removed from the conclusion except for one which a reviewer has requested - Lines 306-318
  2. b) The conclusion has been edited to be brief and concise. We hope this meets your expectations. – Lines 419-432